# AUGMENTED POLICY OPTIMIZATION FOR SAFE REINFORCEMENT LEARNING

## ABSTRACT

Safe reinforcement learning (RL) holds a critical role in acquiring policies that conform to explicit constraints, ensuring their suitability for safety-critical applications. However, methods rooted in the primal-dual concept demonstrate inherent instability. Meanwhile, owing to policy initialization and algorithmic approximation errors, prior methods relying on trust region invariably produce infeasible policies during training, rendering the constructed local optimization problem insoluble. In this paper, we present the Augmented Constraint Policy Optimization (ACPO) algorithm, which encompasses a novel approach to constructing local policy search problems and an optimization problem decomposition method. Specifically, this method introduces an approach for optimizing local policy search that guarantees a solution without relying on hypothetical premises. Utilizing the Alternating Direction Method of Multipliers (ADMM) algorithm as a foundation, we partition the original optimization problem into simpler subproblems that can be efficiently and robustly solved using first-order methods. Comprehensive experimental evaluations have conclusively demonstrated that our proposed method consistently outperforms the baselines in terms of both performance and constraint satisfaction.

## 1 INTRODUCTION

The combination of deep learning (LeCun et al., 2015) and reinforcement learning (RL) (Sutton & Barto, 2018) has ushered in a new era of breakthroughs in diverse domains such as playing Atari games (Mnih et al., 2013; Van Hasselt et al., 2016), Go (Silver et al., 2016; 2017), StarCraft (Team, 2019), robotics (Schulman et al., 2015; 2017; Haarnoja et al., 2018; Singh et al., 2022) and recommendations (Afsar et al., 2022). Nevertheless, the exploration nature of conventional RL methods has hindered their seamless application to real-world challenges (Amodei et al., 2016). This limitation becomes apparent when addressing critical concerns, like safeguarding robots from damage and ensuring human safety during their practical deployment. As a result, the emerging research avenue of safe RL has gained prominence, driven by the pragmatic necessities of our times.

One prevailing approach for addressing such concerns is the adoption of Constrained Markov Decision Processes (CMDP) (Altman, 1999) to model problems, thereby transmuting security considerations into actionable policy set constraints. This extension introduces a challenge: the agents' ability to freely explore their surroundings is curtailed, posing a more formidable learning task than traditional RL. Over the past few years, a plethora of solutions have been proposed. Rooted in the primal-dual method, Tessler et al. (2018) exploited Lagrangian functions to recast constrained conundrums into unconstrained optimization objectives. These frameworks leverage classical RL methodologies to unearth optimal policy. While computationally efficient and conceptually straightforward, they suffer from convergence oscillations and instability issues.

In contrast, methods like Yang et al. (2020) and Yang et al. (2022), which hinged on projection techniques, mitigated the security aspects by embedding projection steps to confine policy within feasible regions. However, their tendency towards overly conservative policies can impede the resolution of practical problems. Building on the foundations of Trust Region Policy Optimization (TRPO) (Schulman et al., 2015), algorithms such as Constrained Policy Optimization (CPO) (Achiam et al., 2017) and First Order Constrained Optimization in Policy Space (FOCOPS) (Zhang et al., 2020) retained the advantages of optimization stability and high-performance outcomes. However, it is

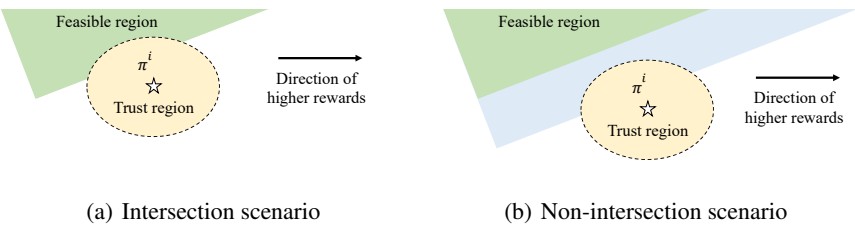



(a) Intersection scenario        (b) Non-intersection scenario

Figure 1: Constrained local policy search



crucial to note that the optimization problem constructed by this method is solvable only when an intersection exists between the trust and the feasible region, as illustrated in Figure 1(a). Conversely, when no such intersection is present, as depicted in Figure 1(b), the constructed optimization problem lacks a feasible solution. Nevertheless, issues like policy initialization and approximation errors, as observed in these methods such as CPO and FOCOPS, present significant challenges that often lead to encountering infeasible optimization problems during training composition. Confronted with this unavoidable scenario, optimizing the policy becomes essentially meaningless.

In this paper, we propose the Augmented Constraint Policy Optimization (ACPO) algorithm attempts to answer the following question: **What is the proper form of local policy search optimization problem for Safe RL?** ACPO introduces a construction method for a more broadly applicable and robust local policy search problem. Specifically, ACPO introduces a novel approach that relaxes the constraints of local policy search, ingeniously extending the feasible region when misalignment with the trust region occurs, as illustrated in Figure 1(b). This guarantees the existence of an optimal solution for the locally defined optimization problem, obviating the need for stringent initialization requirements. Additionally, exact penalty functions (Han & Mangasarian, 1979) are employed to penalize relaxation, thereby ensuring the preservation of the optimal solution's invariance. Furthermore, we employ the Alternating Direction Method of Multipliers (ADMM) (Boyd et al., 2011) algorithm to decompose the optimization problem into multiple unconstrained optimization subproblems. These subproblems alternate and can be efficiently solved using only one step of gradient information.

In summary, this paper makes several significant contributions:

1. *ACPO Algorithm:* We introduce the ACPO algorithm, which features a novel approach to constructing local policy search problems. This approach leverages exact penalty functions to relax constraints, ensuring the solvability of optimization problems. Additionally, we propose an optimization problem decomposition method that breaks down complex constrained optimization problems into unconstrained optimization problems, drawing inspiration from the ADMM algorithm concept.

2. *Theoretical Insights:* We establish the equivalence between ACPO's local policy search construction method and the classical approach, while also offering an upper bound on the cost return during training. Furthermore, we've conducted an analysis of the properties of the algorithm's stable solution.

3. *Superior Performance:* Extensive experimentation validates ACPO's exceptional performance, surpassing several state-of-the-art algorithms in terms of reward improvement and constraint satisfaction while maintaining stable convergence.

## 2 RELATED WORK AND PRELIMINARY

### 2.1 RELATED WORK

The primal-dual approach stands as a widely employed method in solving constrained optimization problems. This approach has given rise to influential algorithms (Chow et al., 2017; Tessler et al., 2018). These algorithms tackle constrained optimization by translating them into unconstrained counterparts through Lagrangian functions. By manipulating the Lagrange multiplier, they ensure the constraints are satisfied. Yet, the primal-dual methods are susceptible to variations in Lagrange

multipliers, rendering optimization unstable. To address this concern, Stooke et al. (2020) drew inspiration from control mechanisms and devised CPPOPID. This method uses a PID controller (Willis, 1999) to update Lagrange multipliers resulting in it converging quickly and stably. Notably, conventional dual methods necessitate separate Lagrange multiplier learning and cannot guarantee constraint satisfaction throughout training.

Apart from the primal-dual paradigm, the projection approach finds prevalence in constrained optimization. Yang et al. (2020; 2022) bifurcated optimization into two phases. Firstly, it exclusively refined the objective function within the trust region, securing the optimal policy. Subsequently, this optimal policy underwent projection into the feasible region. However, the policies derived from such methodologies often display excessive conservatism, not aligning with practical utility. Yang et al. (2021) counteracted this by integrating a baseline policy with outstanding pre-training performance. This policy improvement involves projecting it closer to the baseline policy to ensure performance standards. One drawback of this approach is its sensitivity to the performance of the baseline algorithm, which can impact overall algorithm performance.

In parallel to the aforementioned intricate methodologies, the penalty function method emerges as a straightforward yet effective optimization technique. By directly crafting a penalty function, a minute value prevails within the feasible region while swiftly surging toward infinity as the edge of the region approaches. This category includes notable algorithms such as Interior-point Policy Optimization (IPO) (Liu et al., 2020) employing a logarithmic penalty function, and Penalized Proximal Policy Optimization (P3O) (Zhang et al., 2022) incorporating exact penalty functions. However, the challenge lies in delineating the penalty coefficient magnitude and narrowing the gap with the optimal solution.

Beyond the traditional optimization models outlined earlier, the local policy search approach further extends the unconstrained RL method into constrained optimization (Pirotta et al., 2013; Schulman et al., 2015). CPO (Achiam et al., 2017) ingeniously involved upper and lower bounds from TRPO to transform the primary problem into a local policy search problem. Subsequently, the second-order approximation problem is iteratively solved to secure the policy. While CPO exhibits robust convergence in practical scenarios, the computational costs are substantial due to the information matrix calculations. FOCOPS (Zhang et al., 2020) introduced enhancements over CPO by attaining the approximate optimal solution from the original problem, substantially curtailing computational overhead. It's important to note that this method primarily focuses on analyzing the intersection between feasible region and trust region.

While both our work and the FOCOPS fall under the category of local policy search algorithms, ACPO distinguishes itself by introducing a globally convergent policy optimization approach. ACPO innovatively leverages exact penalty functions to formulate equivalent constrained optimization problems and employs the ADMM (Boyd et al., 2011) algorithm to partition them into multiple unconstrained optimization subproblems, enhancing the overall optimization process. Moreover, we establish a theoretical upper bound for constraint violation when the sampling policy fails to meet the constraints.

## 2.2 Preliminary

A CMDP (Altman, 1999) is formally defined as a tuple $M = (\mathcal{S}, \mathcal{A}, \mathcal{P}, r, \mathcal{C}, \gamma, \rho_0)$, where $\mathcal{S}$ is the state space , $\mathcal{A}$ is the action space, $\mathcal{P}(s'|s, a) : \mathcal{S} \times \mathcal{S} \times \mathcal{A} \to [0, 1]$ is the transition probability function which represents the probability of state transition from $s$ to $s'$ after applying action $a$, $r :$ $\mathcal{S} \times \mathcal{A} \to \mathbb{R}$ is the reward function, $\mathcal{C} = \{(c_k, b_k)\}_{k=1}^K$ is the constraint set (where $c_k : \mathcal{S} \times \mathcal{A} \to \mathbb{R}$ is the $k$-th cost function, and $b_k$ is the $k$-th limits), $\gamma \in (0, 1)$ is the discount factor, and $\rho_0 : \mathcal{S} \to [0, 1]$ is the initial state distribution.

In the context of CMDP, a policy $\pi$ is a probability distribution defined on $\mathcal{S} \times \mathcal{A}$, where $\pi(a|s)$ denotes the probability of selecting action $a$ at state $s$. $\Pi$ denotes the set encompassing all possible policies. Additionally, a stationary parameterized policy $\pi_{\boldsymbol{\theta}}$ is a probability distribution defined on $\mathcal{S} \times \mathcal{A}$, with $\pi_{\boldsymbol{\theta}}(a|s)$ denotes the probability of playing $a$ at state $s$, and $\Pi_{\boldsymbol{\theta}} = \{\pi_{\boldsymbol{\theta}} : \boldsymbol{\theta} \in \mathbb{R}^p\}$ denotes the set of all parameterized policies. Let $d_{\pi}^{s_0}(s) = (1 - \gamma) \sum_{t=0}^{\infty} \gamma^t \mathcal{P}(s_t = s|s_0)$ be the stationary state distribution of the Markov chain starting at state $s_0$. We define $d_{\pi}^{\rho_0}(s) = \mathbb{E}_{s_0 \sim \rho_0}[d_{\pi}^{s_0}(s)]$ as the discounted state visitation distribution on initial distribution $\rho_0$.

We define $\tau = \{s_t, a_t, r(s_t, a_t), c_k(s_t, a_t)\}_{t=0}^{\infty} \sim \pi$ as the trajectory distribution generated by $\pi$, where $s_0 \sim \rho_0$, $a_t \sim \pi(\cdot|s_t)$ and $s_{t+1} \sim \mathcal{P}(\cdot|s_t, a_t)$. We express the state value function of the reward as $V_\pi(s) := \mathbb{E}_{\tau \sim \pi}[\sum_{t=0}^{\infty} \gamma^t r(s_t, a_t)|s_0 = s]$, the state-action value function of the reward as $Q_\pi(s, a) := \mathbb{E}_{\tau \sim \pi}[\sum_{t=0}^{\infty} \gamma^t r(s_t, a_t)|s_0 = s, a_0 = a]$ and the advantage function of the reward is $A_\pi(s, a) := Q_\pi(s, a) - V_\pi(s)$. Similarly, we can define the state value function of the $k$-th cost as $V_\pi^{c_k}(s) := \mathbb{E}_{\tau \sim \pi}[\sum_{t=0}^{\infty} \gamma^t c_k(s_t, a_t)|s_0 = s]$, define the state-action value function of the cost $k$ as $Q_\pi^{c_k}(s, a) := \mathbb{E}_{\tau \sim \pi}[\sum_{t=0}^{\infty} \gamma^t c_k(s_t, a_t)|s_0 = s, a_0 = a]$ and define the advantage function of the cost $k$ as $A_\pi^{c_k}(s, a) := Q_\pi^{c_k}(s, a) - V_\pi^{c_k}(s)$. The expected discount $k$-th cost return is defined as $J_k(\pi) := \mathbb{E}_{s \sim \rho_0}[V_\pi^{c_k}(s)]$ and the expected discount reward return is defined as $J(\pi) := \mathbb{E}_{s \sim \rho_0}[V_\pi(s)]$. The feasible policy set $\Pi^{\mathcal{C}}$ is defined as: $\Pi^{\mathcal{C}} := \bigcap_{k=1}^{K} \{J_k(\pi) \leq b_k\}$. The goal of safe RL is to search the optimal policy $\pi^* = \arg\max_{\pi \in \Pi^{\mathcal{C}}} J(\pi)$.

Solving the CMDP problem directly is challenging and inefficient. Typically, we iteratively update the policy using a fixed policy to gather samples for constructing local optimization problems. Hence, during the $i$-th iteration, we expect to utilize the current policy $\pi^i$ to construct a local optimization problem, resulting in the updated policy $\pi^{i+1}$. Prior research (Peters & Schaal, 2008; Schulman et al., 2015) had demonstrated that the inclusion of local trust domain constraints during the construction of local optimization problems can lead to enhanced efficiency and improved performance. Based on the above work, the CPO algorithm implemented a stringent constraint $\epsilon$ on the Kullback-Leibler (KL) divergence and proposed substituting the original reward and cost return functions with approximate surrogate functions to form a local optimization problem. For simplicity, let $\mathcal{L}_r^{\pi^i}(\pi) := -(1 - \gamma)^{-1} \mathbb{E}_{s \sim d_{\pi^i}^{\rho_0}, a \sim \pi}[A_{\pi^i}(s, a)]$ and $\mathcal{L}_{c_k}^{\pi^i}(\pi) := J_k(\pi^i) + (1 - \gamma)^{-1} \mathbb{E}_{s \sim d_{\pi^i}^{\rho_0}, a \sim \pi}[A_{\pi^i}^{c_k}(s, a)] - b_k$. By incorporating relaxation variables vector $\boldsymbol{\xi}^i = (\xi_1^i, \ldots, \xi_K^i)^T$, the CPO algorithm updates policy as follows:

$$\min_{\pi \in \Pi, \boldsymbol{\xi}^i} \quad \mathcal{L}_r^{\pi^i}(\pi) \tag{1a}$$

$$\text{s.t.} \quad \mathcal{L}_{c_k}^{\pi^i}(\pi) + \xi_k^i = 0, \quad k = 1, \ldots, K \tag{1b}$$

$$\overline{D}_{KL}(\pi||\pi^i) \leq \epsilon \tag{1c}$$

$$\xi_k^i \geq 0, \ k = 1, \ldots, K \tag{1d}$$

where $\overline{D}_{KL}(\pi||\pi^i) := \mathbb{E}_{s \sim d_{\pi^i}^{\rho_0}}[D_{KL}(\pi||\pi^i)[s]]$.

**Definition 1.** *(Slater's condition) There exists a feasible policy $\pi$ within the trust region of the old policy $\pi^i$ : $\overline{D}_{KL}(\pi||\pi^i) \leq \epsilon$.*

However, in contrast to the Markov Decision Process (MDP) (Sutton & Barto, 2018), introducing the KL divergence (Pollard, 2000) hard constraint can lead to potential issues in the CMDP optimization process. Specifically, the feasible regions of (1b), (1c) and (1d) may not intersect, as illustrated in Figure 1. If the updated policy $\pi^i$ violates Slater's condition, the optimization problem (1) becomes infeasible.

## 3 METHOD

### 3.1 FORMULATION OF THE RELAXATION PROBLEM

In this section, we focus on the construction of a novel local policy search optimization problem, aiming to achieve a more tractable formulation while retaining the fundamental essence of the original problem.

Upon analyzing the problem (1), it becomes evident that when the current policy fails to meet Slater's condition, the constraint limits we impose are overly stringent. There is no feasible policy that satisfies the constraint within the KL divergence trust region, resulting in no feasible solution for the problem (1). Therefore, by relaxing the original problem and permitting $\boldsymbol{\xi}$ to assume small negative values, we can always find a feasible solution for the relaxed problem, as illustrated in Figure 1(b). This holds true regardless of whether the provided policy $\pi^i$ adheres to Slater's condition or not. In light of this, we incorporate the exact penalty function method (Han & Mangasarian, 1979), a

penalty function method that does not change the optimal solution, into the problem (1) and rewrite it as follows:

$$\min_{\pi \in \Pi, \boldsymbol{\xi}^i} \quad \mathcal{L}_r^{\pi^i}(\pi) + \sigma g(\boldsymbol{\xi}^i) \tag{2a}$$

$$\text{s.t.} \quad \mathcal{L}_{c_k}^{\pi^i}(\pi) + \xi_k^i = 0, \quad k = 1, \ldots, K \tag{2b}$$

$$\overline{D}_{KL}(\pi || \pi^i) \leq \epsilon \tag{2c}$$

where $\sigma$ is positive penalty parameter and $g(\boldsymbol{\xi}^i) = \sum_{k=1} \max\{0, -\xi_k^i\}$ is exact penalty function. Subsequently, we derive a policy iteration algorithm (Algorithm 1) based on this local policy search optimization problem.

**Theorem 1.** *Assuming $\boldsymbol{v}$ is the optimal Lagrange multipliers to the constraints (1d) of problem (1). Provided that the penalty factor $\sigma$ is a sufficiently large constant ($\sigma \geq ||\boldsymbol{v}||_\infty$), problems (1) and (2) yield the same optimal solution.*

*Proof.* See Appendix A.1.

Theorem 1 establishes the equivalence of the optimal solutions between the relaxed problem (1) and the penalty problem (2). This equivalence holds under the assumption that policy $\pi$ exists within the trust region of the old policy $\pi^i$, satisfying $\overline{D}_{KL}(\pi || \pi^i) \leq \epsilon$. The chosen penalty factor $\sigma$ ensures that both problems converge to the same optimal solution, preserving the integrity and consistency of the optimization process. Furthermore, regardless of whether the current policy $\pi^i$ satisfies Slater's condition or not, we can establish an upper bound on the $k$-th constraint's return for policy $\pi^{i+1}$:

**Proposition 1.** *Suppose $\pi^{i+1}$, $\boldsymbol{\xi}^{i,*}$ are the optimal solution of problem (2), the upper bound on the $k$-th cost of $\pi^{i+1}$ is*

$$J_k(\pi^{i+1}) \leq b_k + \frac{\sqrt{2\epsilon}\gamma\delta_{\pi^{i+1}}^{c_k}}{(1-\gamma)^2} - \xi_k^{i,*} \tag{3}$$

*where $\delta_{\pi^{i+1}}^{c_k} = \max|\mathbb{E}_{a \in \pi^{i+1}}[A_{\pi^i}^{c_k}(s,a)]|$, $\xi_k^{i,*} \in \mathbb{R}$.*

*Proof.* See Appendix A.2.

---

**Algorithm 1** Local search policy iteration algorithm

---

**Input:** Policy $\pi^0$, Penalty parameter $\sigma$.
    **for** $i = 0, 1, \ldots$ until convergence **do**
        Collect a set of trajectories $\mathcal{D}$ with policy $\pi^i$.
        Compute all advantage values $A_{\pi^i}(s,a), \ldots, A_{\pi^i}^{c_K}(s,a)$.
        Construct and solve problem (2) for $\pi^{i+1}, \boldsymbol{\xi}^{i,*}$.
    **end for**
**Output:** $\pi^*, \boldsymbol{\xi}^*$.

---

**Theorem 2.** *With a sufficiently large penalty coefficient $\overline{\sigma}$, the relaxation variable $\boldsymbol{\xi}^*$ in the stable solution produced by Algorithm 1 is guaranteed to be non-negative, and the worst-case constraint violation for the $k$-th constraint in its stable policy $\pi^*$ is:*

$$J_k(\pi^*) \leq b_k + \frac{\sqrt{2\epsilon}\gamma\delta_{\pi^*}^{c_k}}{(1-\gamma)^2} \tag{4}$$

*Proof.* See Appendix A.3.

Theorem 2 investigates the attributes of the algorithm's stable solution. Building on the findings of Theorem 1 and Proposition 1, which analyzes the properties of the solution to the problem (2), Theorem 2 offers further insights into the stable solution generated by Algorithm 1, based on the conclusions drawn earlier. Specifically, Theorem 2 establishes that the relaxation variable $\boldsymbol{\xi}^*$ in the stable solution is consistently non-negative, and the most significant constraint violation in its stable policy aligns with that of the CPO algorithm.

### 3.2 FINDING THE OPTIMAL UPDATE POLICY

The optimization problem (2) involves numerous variables and intricate constraints, posing significant optimization challenges. The ADMM algorithm, rooted in primal-dual augmented Lagrangian problem, offers a decomposition approach that simplifies complex optimization problems into multiple easily solvable subproblems. Employing this method, we break down the original optimization problem into three simpler subproblems. Below, we formulate the original dual augmented Lagrangian problem as follows, with its construction principles detailed in Appendix B.1:

$$\max_{\eta^i \geq 0, \boldsymbol{\lambda}^i} \min_{\pi \in \Pi, \boldsymbol{\xi}^i} L(\pi, \boldsymbol{\lambda}^i, \eta^i, \boldsymbol{\xi}^i; \sigma, \rho) \tag{5}$$

$$L(\pi, \boldsymbol{\lambda}^i, \eta^i, \boldsymbol{\xi}^i; \sigma, \rho) = \mathcal{L}_r^{\pi^i}(\pi) + \eta^i(\overline{D}_{KL}(\pi||\pi^i) - \epsilon) + \sigma g(\boldsymbol{\xi}^i)$$
$$+ \sum_{k=1}^{K} \lambda_k^i(\mathcal{L}_{c_k}^{\pi^i}(\pi) + \xi_k^i) + \sum_{k=1}^{K} \frac{\rho}{2}||\mathcal{L}_{c_k}^{\pi^i}(\pi) + \xi_k^i||^2 \tag{6}$$

where $\rho$ is the proximal factor, $\boldsymbol{\lambda}^i = (\lambda_1^i, \ldots, \lambda_K^i)^T$ and $\eta^i$ are Lagrange multipliers.

For the sake of conciseness, let us introduce the notation $f^i(\pi) := \mathcal{L}_r^{\pi^i}(\pi) + \max_{\eta \geq 0} \eta(\overline{D}_{KL}(\pi||\pi^i) - \epsilon)$. Then, we can process by applying the ADMM (Boyd et al., 2011) algorithm to solve the reformulated problem efficiently, see Appendix B.2 for the ADMM algorithm details. Then we proceed with the following steps in the $j$-th iteration.

- Obtain an optimal policy based on the prior estimation $\boldsymbol{\lambda}^{i,j}$ and $\boldsymbol{\xi}^{i,j}$ by solving the following optimization problem:

$$\pi^{i,j+1} = \arg\min_{\pi \in \Pi} f^i(\pi) + \frac{\rho}{2} \sum_{k=1}^{K} ||\mathcal{L}_{c_k}^{\pi^i}(\pi) + \xi_k^{i,j} + \frac{\lambda_k^{i,j}}{\rho}||_2^2, \tag{7a}$$

- Update $\boldsymbol{\xi}$ by:

$$\boldsymbol{\xi}^{i,j+1} = \arg\min_{\boldsymbol{\xi}} \sigma g(\boldsymbol{\xi}) + \frac{\rho}{2} \sum_{k=1}^{K} ||\mathcal{L}_{c_k}^{\pi^i}(\pi^{i,j+1}) + \xi_k + \frac{\lambda_k^{i,j}}{\rho}||_2^2, \tag{7b}$$

- Update $\boldsymbol{\lambda}$ by:

$$\lambda_k^{i,j+1} = \lambda_k^{i,j} + \rho(\mathcal{L}_{c_k}^{\pi^i}(\pi^{i,j+1}) + \xi_k^{i,j+1}), k = 1, \ldots, K. \tag{7c}$$

**Lemma 1.** *The policy sequence $\{\pi^{i,j}\}_{j=0}^{\infty}$ generated by (7a-7c) converges to $\pi^{i,*}$. Here, $\pi^{i,*}$ represents the optimal solutions of problem (1).*

*Proof.* See Appendix A.4.

Lemma 1 establishes the convergence of the policy sequence $\{\pi^{i,j}\}_{j=0}^{\infty}$ generated by the specified iterative scheme (7a-7c). The sequence converges to the optimal solution $\pi^{i,*}$ of the problem (1), thereby ensuring the attainment of the best feasible policy. The iterative process guarantees that the algorithm progressively approaches the optimal solution as the number of iterations increases, ultimately achieving convergence to the desired policy.

**Theorem 3.** *Given the prior estimation $\boldsymbol{\lambda}^{i,j}$ and $\boldsymbol{\xi}^{i,j}$, the optimal policy $\pi^{i,j+1}$ for problem (7a) takes the form:*

$$\pi^{i,j+1}(a|s) = \frac{\pi^i(a|s)}{Z(s)} \exp\left\{ \frac{A_{\pi^i}(s,a) - \sum_{k=1}^{K}[\lambda_k^{i,j} + \rho(\mathcal{L}_{c_k}^{\pi^i}(\pi) + \xi_k^{i,j})]A_{\pi^i}^{c_k}(s,a)}{\eta^{i,j+1}(1-\gamma)} \right\} \tag{8}$$

*where $Z(s)$ serves as a constant normalizer that ensures $\pi$ belongs to the policy set $\Pi$ and the dual variables $\eta^{i,j+1}$ represent the solutions to the following convex optimization problem:*

$$\eta^{i,j+1} = \max_{\eta \geq 0} L(\pi^{i,j+1}, \boldsymbol{\lambda}^{i,j}, \eta, \boldsymbol{\xi}^{i,j}; \sigma, \rho) \tag{9}$$

*Proof.* See Appendix A.5.

Through observing the results of Theorem 3, we discover that solving the problem (7a) optimally is akin to addressing an unconstrained problem by maximizing a weighted average of the reward and cost. We identify the constraint coefficient as a composition of error accumulation terms $\lambda_k^{i,j}$, error terms $\rho(\mathcal{L}_{c_k}^{\pi^i}(\pi^i) + \xi_k^{i,j})$, and differential terms $\rho(\mathcal{L}_{c_k}^{\pi^i}(\pi) - \mathcal{L}_{c_k}^{\pi^i}(\pi^i))$. This insight reveals that our algorithm implicitly incorporates a PID controller for learning Lagrange multipliers, enhancing learning stability.

### 3.3 PRACTICAL IMPLEMENTATION

While solving the problem (7a), allowing $\pi$ to reside within the policy space $\Pi$ may result in a policy that does not necessarily belong to the parameterized policy space $\Pi_{\boldsymbol{\theta}}$. Consequently, evaluating or sampling from $\pi$ may no longer be feasible. To address this issue, we replace the policy $\pi$ with the parameterized policy $\pi_{\boldsymbol{\theta}}$.

To bolster computational efficiency and simplify complexity, our proposed algorithm in this paper embraces a first-order approach. Specifically, we utilize the Proximal Policy Optimization (PPO) Algorithm (Schulman et al., 2017) to optimize the policy, incorporating a line search approach and advantage function clipping to ensure adherence to the KL divergence constraint. For a comprehensive understanding of the ACPO algorithm, please refer to Algorithm 2. Detailed descriptions can be found in Appendix D. The loss function of the ACPO algorithm is expressed as follows:

$$Loss(\boldsymbol{\theta}) = -\frac{1}{1-\gamma} \mathbb{E}_{s \sim d_{\pi_{\boldsymbol{\theta}^i}}^{\rho_0}, a \sim \pi_{\boldsymbol{\theta}^i}} [\min\{r^i(\boldsymbol{\theta}) A_{\pi_{\boldsymbol{\theta}^i}}(s,a), \text{clip}(r^i(\boldsymbol{\theta}), 1-\varepsilon, 1+\varepsilon) A_{\pi_{\boldsymbol{\theta}^i}}(s,a)\}]$$

$$+ \frac{\rho}{2} \sum_{k=1}^{K} \|\frac{1}{1-\gamma} \max\{r^i(\boldsymbol{\theta}) A_{\pi_{\boldsymbol{\theta}^i}}(s,a), \text{clip}(r^i(\boldsymbol{\theta}), 1-\varepsilon, 1+\varepsilon) A_{\pi_{\boldsymbol{\theta}^i}}(s,a)\} + \xi_k^{i,j} + \frac{\lambda_k^{i,j}}{\rho}\|_2^2 \tag{10}$$

where $r^i(\boldsymbol{\theta}) = \frac{\pi_{\boldsymbol{\theta}}(a|s)}{\pi_{\boldsymbol{\theta}^i}(a|s)}$ is the importance sampling ratio.

---

**Algorithm 2** ACPO Outline

---

**Input:** Policy network $\pi_{\boldsymbol{\theta}^0}$, Value networks $V_{\phi^0}, V_{\phi_1^0}^{c_1}, \ldots, V_{\phi_K^0}^{c_K}$.

  **while** stopping criteria not meet **do**
    Generate trajectories $\tau \sim \pi_{\boldsymbol{\theta}^i}$.
    Estimate returns and advantage functions.
    **for** each iteration **do**
      **for** each minibatch **do**
        Update value networks by minimizing MSE of $V_\phi, V_\phi^{target}, \ldots, V_{\phi_K}^{c_K}, V_{\phi_K}^{c_K, target}$.
        Update policy network minimizing problem (10).
      **end for**
      **if** $\overline{D}_{KL}(\pi_{\boldsymbol{\theta}} || \pi_{\boldsymbol{\theta}^i}) > \epsilon$ **then**
        Break
      **end if**
      Update $\boldsymbol{\xi}^i$ by minimizing problem (7b).
      Update $\boldsymbol{\lambda}^i$ using equation (7c).
    **end for**
  **end while**

---

## 4 EXPERIMENTS

### 4.1 TASKS

Safety-gymnasium (Ji et al., 2023) integrated and developed a Gym environment by amalgamating environments like Bullet-safety-gym (Gronauer, 2022) and MuJoCo (Todorov et al., 2012), thereby

offering a comprehensive set of RL tasks focused on safety and security. We have crafted three unique experimental scenarios, each featuring varying levels of difficulty coefficients. Further details about these experiments are outlined in Appendix C, and you can access the code on our GitHub repository: https://github.com/SDsly/ACPO.

**Safe Velocity**  This task imposes a safety constraint on the agent's speed in Mujoco, capping it at 50% of the agent's speed after one million steps of optimization using PPO algorithms. We employ four distinct agents, each with its own configuration: *SafetyAnt*, *SafetyHalfCheetah*, *SafetyHopper* and *SafetyHumanoid*. See Appendix C.1 for more details.

**Safe Navigation**  Safe navigation tasks demand agents to interact with the environment, achieving specific goals while maintaining their own safety and avoiding damage to other objects in the environment. In this study, we define two motion modes: *Circle* and *Goal* and employ *Point* and *Car* agents to complete tasks in these respective modes, labeled *PointCircle*, *CarCircle*, *PointGoal*, and *CarGoal*. See Appendix C.2 for more details.

## 4.2  RESULTS

**Baseline Algorithms**  To assess the efficacy of our proposed feasible region relaxation method in enhancing the performance of local policy search algorithms, we conducted experiments using CPO (Achiam et al., 2017) and FOCOPS (Zhang et al., 2020) as our baseline algorithms. Furthermore, to evaluate the impact of optimization stability achieved through the ADMM algorithm, we introduced CPPOPID (Stooke et al., 2020) to our baseline. Additionally, to investigate the distinction between directly employing penalty functions for constraint handling, we included the P3O (Zhang et al., 2022) algorithm as another baseline in our comparative analysis.

**Comparison to baselines**  Figures 3 and 2 illustrate that, in the majority of tasks, ACPO outperforms other baseline algorithms in terms of rewards while adhering to cost constraints. Moreover, ACPO exhibits smoother convergence in its training curves and lower variance. Notably, in the *SafetySwimmer* task, although the average reward performance of ACPO is slightly lower than CP-POPID, it is imperative to highlight that CPPOPID's training curves exhibit extreme variance and heavy reliance on initialization.

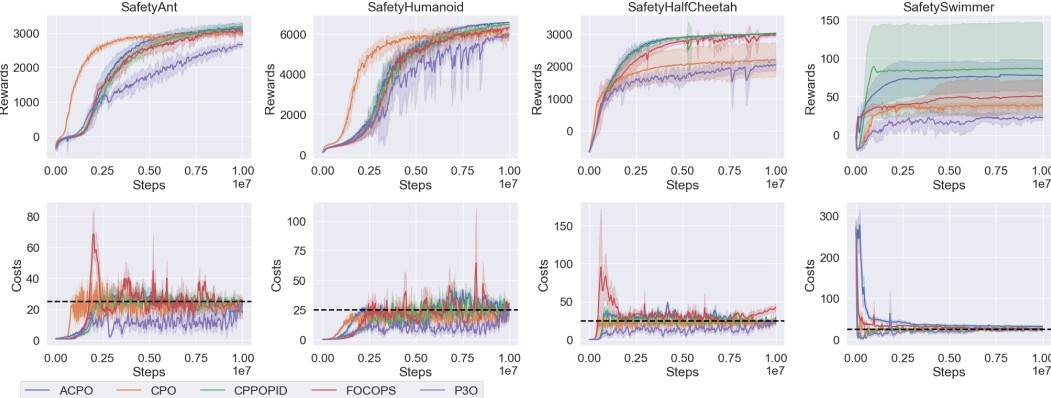

Figure 2: Training curves in Safe Velocity. The rewards and costs are obtained from 10 million interaction steps, with dashed black lines indicating the target cost value set at 25.

In comparison to closely related local policy search methods like CPO and FOCOPS, FOCOPS demonstrates unstable convergence, coupled with significant early-phase cost fluctuations and higher variance in tasks such as *Goal*. Conversely, CPO exhibits stable convergence and strong constraint satisfaction performance, but it occasionally exhibits suboptimal reward performance in certain tasks and necessitates second-order information, incurring substantial computational overhead. Despite both ACPO and P3O employing exact penalty functions for constraint violations, P3O consistently lags behind other algorithms in terms of reward performance, especially in tasks like *Goal*. In a comprehensive assessment, ACPO stands out for its simplicity of implementation, stable

convergence, and superior performance. However, it's important to note that ACPO does exhibit slower convergence in cost for complex problems, particularly in the *Goal* environment.

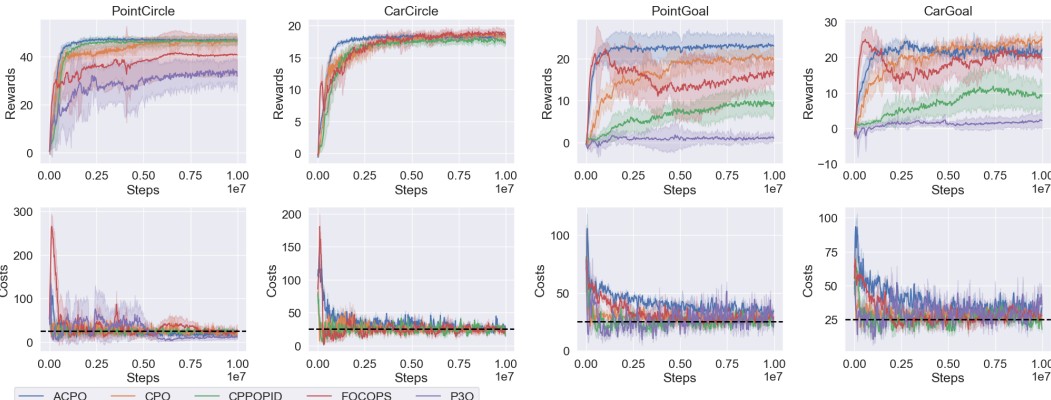

Figure 3: Training curves for safety navigation. The X-axis represents the number of samples used, while the Y-axis represents the average total reward or cost return over the last 100 episodes. Dashed black lines indicate the target cost value set at 25.

**Generalization Analysis**   To assess the algorithm's performance, we followed the methodology outlined by Mnih et al. (2015). We initially trained our model using a fixed random seed and subsequently conducted 100 trials using 10 different random seeds for testing. The performance of each algorithm in these trials is summarized in Tables 1. Notably, ACPO consistently demonstrates the ability to satisfy constraints across the majority of environments and, with the exception of the *Goal* environment, achieves optimal performance.

Table 1: Performance evaluation

| Environment | | CPO | FOCOPS | CPPOPID | P3O | **ACPO** |
|---|---|---|---|---|---|---|
| PointCircle | Reward | $92.03 \pm 7.67$ | $94.19 \pm 5.22$ | $94.60 \pm 2.09$ | $56.06 \pm 1.70$ | $\mathbf{94.18 \pm 2.58}$ |
| | Cost(<25) | $15.02 \pm 59.23$ | $84.12 \pm 93.12$ | $46.91 \pm 34.26$ | $0.97 \pm 6.20$ | $13.70 \pm 23.15$ |
| CarCircle | Reward | $37.32 \pm 1.76$ | $37.87 \pm 2.65$ | $35.58 \pm 1.40$ | $37.32 \pm 1.76$ | $\mathbf{34.59 \pm 1.69}$ |
| | Cost(<25) | $64.67 \pm 51.10$ | $92.09 \pm 54.28$ | $35.03 \pm 35.16$ | $64.67 \pm 51.10$ | $16.11 \pm 31.04$ |
| PointGoal | Reward | $\mathbf{19.98 \pm 4.21}$ | $16.72 \pm 9.07$ | $9.44 \pm 5.74$ | $0.82 \pm 2.52$ | $20.51 \pm 4.86$ |
| | Cost(<25) | $26.45 \pm 26.08$ | $29.28 \pm 42.61$ | $28.62 \pm 52.08$ | $27.94 \pm 95.18$ | $30.71 \pm 30.37$ |
| CarGoal | Reward | $23.60 \pm 7.54$ | $19.76 \pm 9.12$ | $\mathbf{12.15 \pm 6.93}$ | $-0.10 \pm 1.90$ | $21.95 \pm 6.42$ |
| | Cost(<25) | $30.33 \pm 31.96$ | $30.83 \pm 42.16$ | $24.62 \pm 36.53$ | $15.74 \pm 49.27$ | $31.36 \pm 31.40$ |
| SafetyAnt | Reward | $3030.91 \pm 197.32$ | $3035.05 \pm 514.98$ | $3270.54 \pm 297.76$ | $2790.09 \pm 29.87$ | $\mathbf{3294.34 \pm 16.37}$ |
| | Cost(<25) | $14.32 \pm 5.72$ | $13.87 \pm 6.18$ | $29.97 \pm 23.19$ | $7.14 \pm 4.39$ | $18.64 \pm 7.64$ |
| SafetyHalfCheetah | Reward | $1844.11 \pm 19.27$ | $2957.40 \pm 19.14$ | $3002.13 \pm 145.33$ | $1898.66 \pm 21.45$ | $\mathbf{3005.01 \pm 5.51}$ |
| | Cost(<25) | $20.96 \pm 5.79$ | $0.50 \pm 0.77$ | $2.18 \pm 1.92$ | $27.83 \pm 6.88$ | $5.14 \pm 2.22$ |
| SafetySwimmer | Reward | $39.49 \pm 1.48$ | $44.63 \pm 0.95$ | $41.26 \pm 2.87$ | $21.13 \pm 11.61$ | $\mathbf{74.81 \pm 1.32}$ |
| | Cost(<25) | $26.67 \pm 1.48$ | $25.19 \pm 1.59$ | $23.16 \pm 7.29$ | $42.92 \pm 45.81$ | $23.03 \pm 1.94$ |
| SafetyHumanoid | Reward | $6388.03 \pm 5.64$ | $6549.05 \pm 7.38$ | $6605.81 \pm 8.53$ | $6355.88 \pm 15.53$ | $\mathbf{6657.39 \pm 4.11}$ |
| | Cost(<25) | $0.16 \pm 0.31$ | $20.44 \pm 19.76$ | $36.50 \pm 50.27$ | $285.78 \pm 63.10$ | $19.60 \pm 21.56$ |

## 5   DISCUSSION

In this work, we introduce a novel method for constructing constrained local policy search problems without the need for initial solutions. Building upon this method, we present ACPO, a security RL algorithm that is characterized by its simplicity, computational efficiency, and effectiveness. We are confident that our proposed method holds substantial potential and can deliver enhanced performance through refinements in policy optimization. Furthermore, our algorithm possesses decomposition characteristics, making it well-suited for expansion into the realm of multi-agent systems and the resolution of increasingly complex challenges.

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

# A  PROOFS

## A.1  PROOF OF THEOREM 1

**Lemma 2.**    *Let $\hat{\pi}$ and $\hat{\boldsymbol{\xi}}$ represent the optimal solution of problem (1), and let $\boldsymbol{\upsilon}$ be the optimal Lagrange multipliers for constraints (1d). When the penalty factor $\sigma$ is chosen as a sufficiently large constant ($\sigma \geq ||\boldsymbol{\upsilon}||_\infty$), policy $\hat{\pi}$ also qualifies as an optimal solution for problem (2).*

*Proof.* Given any policy $\pi$ and relaxation variables $\boldsymbol{\xi}$ that satisfy the constraint conditions of problem (2), it follows that:

$$\mathcal{L}_r^{\pi^i}(\pi) + \sigma g(\boldsymbol{\xi}) = \mathcal{L}_r^{\pi^i}(\pi) + \sigma \sum_{k=1}^{K} \max\{0, -\xi_k\} \tag{11}$$

$$\geq \mathcal{L}_r^{\pi^i}(\pi) + \sum_{k=1}^{K} \mu_k \max\{0, -\xi_k\} \tag{12}$$

$$\geq \mathcal{L}_r^{\pi^i}(\pi) + \sum_{k=1}^{K} -\mu_k \xi_k \tag{13}$$

$$\geq \mathcal{L}_r^{\pi^i}(\pi) + \sum_{k=1}^{K} \lambda_k (\mathcal{L}_{c_k}^{\pi^i}(\pi) - \xi_k) + \sum_{k=1}^{K} -\mu_k \xi_k + \eta(\overline{D}_{KL}(\pi||\pi^i) - \epsilon) \tag{14}$$

$$\geq \mathcal{L}_r^{\pi^i}(\hat{\pi}) + \sum_{k=1}^{K} \lambda_k (\mathcal{L}_{c_k}^{\pi^i}(\hat{\pi}) - \hat{\xi}_k) + \sum_{k=1}^{K} -\mu_k \hat{\xi}_k + \eta(\overline{D}_{KL}(\hat{\pi}||\pi^i) - \epsilon) \tag{15}$$

$$\geq \mathcal{L}_r^{\pi^i}(\hat{\pi}) + \sum_{k=1}^{K} -\mu_k \hat{\xi}_k \tag{16}$$

$$= \mathcal{L}_r^{\pi^i}(\hat{\pi}) + \sum_{k=1}^{K} \mu_k \max\{0, -\hat{\xi}_k\} \tag{17}$$

$$= \mathcal{L}_r^{\pi^i}(\hat{\pi}) + \sigma \sum_{k=1}^{K} \max\{0, -\hat{\xi}_k\} \tag{18}$$

$$= \mathcal{L}_r^{\pi^i}(\hat{\pi}) + \sigma g(\hat{\boldsymbol{\xi}}) \tag{19}$$

Under the condition $\sigma \geq ||\boldsymbol{\upsilon}||_\infty$, equation (12) is verified to be true. Moreover, the fulfillment of constraints (2b) and (2c) by policy $\pi$ and relaxation variables $\boldsymbol{\xi}$ ensures the validity of equation (14). Equation (15) holds since $\hat{\pi}$ and $\hat{\boldsymbol{\xi}}$ minimize the Lagrange function. Additionally, equation (18) is derived from the complementary slackness condition.

By considering equations (11) to (19), we can conclude that $\hat{\pi}$ is a minimizer of the problem (2). The proof of Lemma 2 is thereby completed.

**Lemma 3.**    *Given $\tilde{\pi}$ and $\tilde{\boldsymbol{\xi}}$ represent the optimal solution for problem (2), and let $\boldsymbol{\mu}$ be the optimal Lagrange multipliers for constraints (1d). Assuming that the penalty factor $\sigma$ is sufficiently large ($\sigma \geq ||\boldsymbol{\upsilon}||_\infty$), policy $\tilde{\pi}$ is also an optimal solution for problem (1).*

*Proof.* Given any policy $\pi$ and relaxation variables $\boldsymbol{\xi}$ that satisfy the constraint conditions of problem (1), it follows that:

$$\mathcal{L}_r^{\pi^i}(\tilde{\pi}) \tag{20}$$

$$\leq \mathcal{L}_r^{\pi^i}(\tilde{\pi}) + \sigma \sum_{k=1}^{K} \max\{0, -\tilde{\xi}_k\} \tag{21}$$

$$\leq \mathcal{L}_r^{\pi^i}(\pi) + \sigma \sum_{k=1}^{K} \max\{0, -\xi_k\} \tag{22}$$

$$= \mathcal{L}_r^{\pi^i}(\pi) \tag{23}$$

Equation (22) holds true since both policy $\tilde{\pi}$ and relaxation variables $\tilde{\boldsymbol{\xi}}$ minimize the objective function.

**Theorem 1.** *Assuming $\boldsymbol{v}$ is the optimal Lagrange multipliers to the constraints (1d) of problem (1). Provided that the penalty factor $\sigma$ is a sufficiently large constant ($\sigma \geq ||\boldsymbol{v}||_\infty$), problems (1) and (2) yield the same optimal solution.*

*Proof.* Based on the results of Lemma 2 and Lemma 3, Theorem 1 is proven, demonstrating the equivalence of optimal solutions between problems (1) and (2) when the specified conditions are met.

### A.2 PROOF OF PROPOSITION 1

**Lemma 4.** *For any policies $\pi'$, $\pi$, and any cost function $c_k$, with $\delta_\pi^{c_k} = \max|\mathbb{E}_{a\in\pi'}[A_\pi^{c_k}(s,a)]|$, the following bounds hold:*

$$J_k(\pi') - J_k(\pi) \leq \frac{1}{1-\gamma}\mathbb{E}_{s\sim d_\pi^{\rho_0}, a\sim\pi'}\left[A_\pi^{c_k}(s,a) + \frac{2\gamma\delta_{\pi'}^{c_k}}{1-\gamma}D_{TV}(\pi'||\pi)[s]\right] \tag{24a}$$

$$\leq \frac{1}{1-\gamma}\mathbb{E}_{s\sim d_\pi^{\rho_0}, a\sim\pi'}\left[A_\pi^{c_k}(s,a)\right] + \frac{2\gamma\delta_{\pi'}^{c_k}}{(1-\gamma)^2}\sqrt{\frac{1}{2}\mathbb{E}_{s\sim d_\pi^{\rho_0}, a\sim\pi'}[D_{KL}(\pi'||\pi)[s]]} \tag{24b}$$

$$\leq \frac{1}{1-\gamma}\mathbb{E}_{s\sim d_\pi^{\rho_0}, a\sim\pi'}\left[A_\pi^{c_k}(s,a)\right] + \frac{\sqrt{2\epsilon}\gamma\delta_{\pi'}^{c_k}}{(1-\gamma)^2} \tag{24c}$$

*Proof.* The derivation of (24a) is achieved by applying Corollary 2 of Achiam et al. (2017), while (24b) is obtained through Corollary 3.

**Proposition 1.** *Suppose $\pi^{i+1}$, $\boldsymbol{\xi}^{i,*}$ is the optimal solution of problem (2), the upper bound on the $k$-th return of $\pi^{i+1}$ is:*

$$J_k(\pi^{i+1}) \leq b_k + \frac{\sqrt{2\epsilon}\gamma\delta_{\pi^{i+1}}^{c_k}}{(1-\gamma)^2} - \xi_k^{i,*} \tag{25}$$

*where $\delta_{\pi^{i+1}}^{c_k} = \max|\mathbb{E}_{a\in\pi^{i+1}}[A_{\pi^i}^{c_k}(s,a)]|$.*

*Proof.* Let's begin by considering the policy definitions $\pi^{i+1}$ and $\boldsymbol{\xi}^{i,*}$:

$$J_k(\pi^i) + \frac{1}{1-\gamma}\mathbb{E}_{s\sim d_{\pi^i}^{\rho_0}, a\sim\pi^{i+1}}[A_{\pi^i}^{c_k}(s,a)] + \xi_k^{i,*} = b_k \tag{26}$$

Substituting equation (26) into inequality (24c) results in:

$$J_k(\pi^{k+1}) + \frac{1}{1-\gamma}\mathbb{E}_{s\sim d_{\pi^i}^{\rho_0}, a\sim\pi^{i+1}}[A_{\pi^i}^{c_k}(s,a)] - b_k + \xi_k^{i,*} \leq$$

$$\frac{1}{1-\gamma}\mathbb{E}_{s\sim d_\pi^{\rho_0}, a\sim\pi^{i+1}}\left[A_{\pi^i}^{c_k}(s,a)\right] + \frac{\sqrt{2\epsilon}\gamma\delta_{\pi^{i+1}}^{c_k}}{(1-\gamma)^2} \tag{27}$$

$$J_k(\pi^{i+1}) \leq b_k + \frac{\sqrt{2\epsilon}\gamma\delta_{\pi^{i+1}}^{c_k}}{(1-\gamma)^2} - \xi_k^{i,*} \tag{28}$$

This concludes the proof.

## A.3 PROOF OF THEOREM 2

We define the policy set $\Pi^{\pi^i}(\boldsymbol{\xi})$, the domain $\Xi^{\pi^i}$ and the function $l^{\pi^i}(\boldsymbol{\xi})$ as:

$$\Pi^{\pi^i}(\boldsymbol{\xi}) := \{\pi | \mathcal{L}_{c_k}^{\pi^i}(\pi) + \xi_k^i = 0, \overline{D}_{KL}(\pi||\pi^i) - \epsilon \le 0\} \tag{29}$$

$$\Xi^{\pi^i} := \{\boldsymbol{\xi} | \boldsymbol{\xi} = \boldsymbol{\mathcal{L}}^{\pi^i}(\pi), \pi \text{ such that } \overline{D}_{KL}(\pi||\pi^i) - \epsilon \le 0\} \tag{30}$$

$$l^{\pi^i}(\boldsymbol{\xi}) = \inf_{\pi \in \Pi(\boldsymbol{\xi})} \mathcal{L}_r^{\pi^i}(\pi) \tag{31}$$

where $\boldsymbol{\mathcal{L}}^{\pi^i}(\pi) = (\mathcal{L}_{c_1}^{\pi^i}(\pi), \dots, \mathcal{L}_{c_K}^{\pi^i}(\pi))^T$.

**Lemma 5.** $l^{\pi^i}(\boldsymbol{\xi})$ is a convex function, the domain of $\boldsymbol{\xi}$ is $\Xi^{\pi^i}$.

*Proof.* To prove that $\Xi^{\pi^i}$ is a convex set, we begin by considering two arbitrary points in $\Xi^{\pi^i}$, denoted as $\boldsymbol{\mu}$ and $\boldsymbol{\nu}$, satisfying the following conditions:

$$\boldsymbol{\mu} = \boldsymbol{\mathcal{L}}^{\pi^i}(\pi^{\boldsymbol{\mu}}) \tag{32}$$

$$\overline{D}_{KL}(\pi^{\boldsymbol{\mu}}||\pi^i) \le \epsilon \tag{33}$$

$$\boldsymbol{\nu} = \boldsymbol{\mathcal{L}}^{\pi^i}(\pi^{\boldsymbol{\nu}}) \tag{34}$$

$$\overline{D}_{KL}(\pi^{\boldsymbol{\nu}}||\pi^i) \le \epsilon \tag{35}$$

Now, for any scalar $\psi \in [0,1]$, we consider the convex combination of $\boldsymbol{\mu}$ and $\boldsymbol{\nu}$. Leveraging the linearity property of the function $\boldsymbol{\mathcal{L}}^{\pi^i}$, we obtain:

$$\psi\boldsymbol{\mu} + (1-\psi)\boldsymbol{\nu} = \psi\boldsymbol{\mathcal{L}}^{\pi^i}(\pi^{\boldsymbol{\mu}}) + (1-\psi)\boldsymbol{\mathcal{L}}^{\pi^i}(\pi^{\boldsymbol{\nu}}) \tag{36}$$

$$= \boldsymbol{\mathcal{L}}^{\pi^i}(\psi\pi^{\boldsymbol{\mu}} + (1-\psi)\pi^{\boldsymbol{\nu}}) \tag{37}$$

By the convexity of KL divergence, we can write:

$$\overline{D}_{KL}(\psi\pi^{\boldsymbol{\mu}} + (1-\psi)\pi^{\boldsymbol{\nu}}||\pi^i) \le \psi\overline{D}_{KL}(\pi^{\boldsymbol{\mu}}||\pi^i) + (1-\psi)\overline{D}_{KL}(\pi^{\boldsymbol{\nu}}||\pi^i) \tag{38}$$

$$\le \psi\epsilon + (1-\psi)\epsilon \tag{39}$$

$$\le \epsilon \tag{40}$$

This inequality shows that the convex combination of $\psi\boldsymbol{\mu} + (1-\psi)\boldsymbol{\nu}$ belongs to $\Xi^{\pi^i}$. According to the definition of convex sets, we can prove that the feasible domain $\Xi^{\pi^i}$ is a convex set.

Then, we prove that the function $l^{\pi^i}(\boldsymbol{\xi})$ is a convex function. For any $\alpha \in [0,1]$ and $\boldsymbol{x}, \boldsymbol{y} \in \mathbb{R}^K$, given $\pi_1 \in \Pi(\boldsymbol{x})$ and $\pi_2 \in \Pi(\boldsymbol{y})$, there exists $\alpha\pi_1 + (1-\alpha)\pi_2 \in \Pi(\alpha\boldsymbol{x} + (1-\alpha)\boldsymbol{y})$.

From the given definitions, we have the following conditions:

$$\mathcal{L}_{c_k}^{\pi^i}(\pi_1) + x_k = 0 \tag{41}$$

$$\overline{D}_{KL}(\pi_1||\pi^i) - \epsilon \le 0 \tag{42}$$

$$\mathcal{L}_{c_k}^{\pi^i}(\pi_2) + y_k = 0 \tag{43}$$

$$\overline{D}_{KL}(\pi_2||\pi^i) - \epsilon \le 0 \tag{44}$$

As the KL divergence function is convex, we can apply the convex combination property:

$$\overline{D}_{KL}(\alpha\pi_1 + (1-\alpha)\pi_2||\pi^i)$$
$$\le \alpha\overline{D}_{KL}(\pi_1||\pi^i) + (1-\alpha)\overline{D}_{KL}(\pi_2||\pi^i)$$
$$\le \epsilon \tag{45}$$

We add (41) that is $\alpha$ times larger and (42) that is $(1 - \alpha)$ times larger to obtain:

$$\alpha(\mathcal{L}_{c_k}^{\pi^i}(\pi_1) + x_k) + (1 - \alpha)(\mathcal{L}_{c_k}^{\pi^i}(\pi_2) + y_k) = 0 \tag{46}$$

$$\alpha\mathcal{L}_{c_k}^{\pi^i}(\pi_1) + (1 - \alpha)\mathcal{L}_{c_k}^{\pi^i}(\pi_2) + \alpha x_k + (1 - \alpha)y_k = 0 \tag{47}$$

$$\mathcal{L}_{c_k}^{\pi^i}(\alpha\pi_1 + (1 - \alpha)\pi_2) + \alpha x_k + (1 - \alpha)y_k = 0 \tag{48}$$

Thus, we have shown that $\alpha\pi_1 + (1 - \alpha)\pi_2 \in \Pi(\alpha\boldsymbol{x} + (1 - \alpha)\boldsymbol{y})$.

For any $\beta \in [0, 1]$ and $\boldsymbol{x}, \boldsymbol{y} \in \mathbb{R}^K$, suppose $\pi_1 = \underset{\pi \in \Pi(\boldsymbol{x})}{\operatorname{argmin}} \mathcal{L}_r^{\pi^i}(\pi)$ and $\pi_2 = \underset{\pi \in \Pi(\boldsymbol{y})}{\operatorname{argmin}} \mathcal{L}_r^{\pi^i}(\pi)$. Then, we have:

$$l^{\pi^i}(\beta\boldsymbol{x} + (1 - \beta)\boldsymbol{y}) = \inf_{\pi \in \Pi(\beta\boldsymbol{x} + (1 - \beta)\boldsymbol{y})} \mathcal{L}_r^{\pi^i}(\pi) \tag{49}$$

$$\leq \mathcal{L}_r^{\pi^i}(\beta\pi_1 + (1 - \beta)\pi_2) \tag{50}$$

$$\leq \beta\mathcal{L}_r^{\pi^i}(\pi_1) + (1 - \beta)\mathcal{L}_r^{\pi^i}(\pi_2) \tag{51}$$

$$= \beta l^{\pi^i}(\pi_1) + (1 - \beta)l^{\pi^i}(\pi_2) \tag{52}$$

(50) holds due to the definition of the operator $\inf$, (51) follows from the Jensen's inequality, and (52) is true based on the definition of the policies $\pi_1$ and $\pi_2$. This proves that $l^{\pi^i}(\boldsymbol{\xi})$ is a convex function.

**Theorem 2.** *With a sufficiently large penalty coefficient $\overline{\sigma}$, the relaxation variable $\boldsymbol{\xi}^*$ in the stable solution produced by Algorithm 1 is guaranteed to be non-negative, and the worst-case constraint violation for the $k$-th constraint in its stable policy $\pi^*$ is:*

$$J_k(\pi^*) \leq b_k + \frac{\sqrt{2\epsilon}\gamma\delta_{\pi^*}^{c_k}}{(1 - \gamma)^2} \tag{53}$$

*Proof.* Assuming that the optimization process converges to a stable solution after the $z$-th iteration, we can deduce that $\pi^z$ and $\boldsymbol{\xi}^{z,*}$ represent the optimal solutions for the following problems:

$$\min_{\pi \in \Pi, \boldsymbol{\xi}^z} \quad \mathcal{L}_r^{\pi^z}(\pi) + \sigma^z g(\boldsymbol{\xi}^z) \tag{54}$$

$$\text{s.t.} \quad \mathcal{L}_{c_k}^{\pi^z}(\pi) + \xi_k^z = 0, \quad k = 1, \ldots, K \tag{55}$$

$$\overline{D}_{KL}(\pi||\pi^z) \leq \epsilon \tag{56}$$

Then, we can rewrite the above problem as:

$$\min_{\boldsymbol{\xi} \in \Xi^{\pi^z}} \quad l^{\pi^z}(\boldsymbol{\xi}) + \sigma^z g(\boldsymbol{\xi}) \tag{57}$$

As function $\mathcal{L}_{c_k}^{\pi^z}(\pi)$ is linear, it is evident that the extreme value of $\mathcal{L}_{c_k}^{\pi^z}(\pi)$ must reside on the boundary of the feasible region. Hence, $\boldsymbol{\xi}^{z,*}$ must fall within the domain $\Xi^{\pi^z}$. Then, leveraging the Karush-Kuhn-Tucker (KKT) conditions, we can derive the following insights:

$$\nabla l^{\pi^z}(\boldsymbol{\xi}^{z,*}) + \sigma^z \nabla g(\boldsymbol{\xi}^{z,*}) = 0 \tag{58}$$

By definition, we know that:

$$\nabla g(\boldsymbol{\xi})[k] = \begin{cases} 0, & \xi_k > 0 \\ [-1, 0], & \xi_k = 0 \\ -1, & \xi_k < 0 \end{cases} \tag{59}$$

Then, if we give a new penalty parameter $\hat{\sigma} > \sigma^z$ and we can get:

$$\nabla l^{\pi^z}(\boldsymbol{\xi}^{z,*}) + \hat{\sigma} \nabla g(\boldsymbol{\xi}^{z,*}) \leq 0 \tag{60}$$

The inequality (60) holds true with an equality sign if and only if $\boldsymbol{\xi}^{z,*} \geq 0$. Since the function $l^{\pi^i}$ is a convex function, it can be known that the second derivative of $l_{\pi^i}(\boldsymbol{\xi}^{z,*})$ is a positive-definite matrix, and the gradient is monotonically increased, so the optimal solution $\hat{\boldsymbol{\xi}}^{z,*}$ for $\hat{\sigma}$:

$$\hat{\boldsymbol{\xi}}^{z,*} \geq \boldsymbol{\xi}^{z,*} \tag{61}$$

Hence, we can draw the conclusion that by introducing a larger penalty coefficient, we can effectively mitigate the extent of policy constraint violations. Building upon the preceding analysis, we can deduce that there always exists a penalty coefficient, denoted as $\overline{\sigma} > \max\{||\nabla l(\boldsymbol{\xi})||_\infty\}$, ensuring that the stable solution $\boldsymbol{\xi}^*$ satisfies the condition $\boldsymbol{\xi}^* \geq 0$. From the preceding analysis, it can be concluded that $g(\boldsymbol{\xi}^*)$ monotonically decreases and converges to 0. Then, employing Proposition 1, we can derive an upper bound on the $k$-th cost return:

$$J_k(\pi^*) \leq b_k + \frac{\sqrt{2\epsilon}\gamma\delta_{\pi^*}^{c_k}}{(1-\gamma)^2} - \xi_k^* \tag{62}$$

$$\leq b_k + \frac{\sqrt{2\epsilon}\gamma\delta_{\pi^*}^{c_k}}{(1-\gamma)^2} \tag{63}$$

### A.4 Proof of Lemma 1

**Lemma 1.** *The policy sequence $\{\pi^{i,j}\}_{j=0}^\infty$ generated by (7a-7c) converges to $\pi^{i,*}$. Here, $\pi^{i,*}$ represent the optimal solutions of problem (1).*

*Proof.* By leveraging the previously defined notation and incorporating the augmented Lagrangian primal-dual method with a quadratic proximal operator, we can transform the problem into the following form:

$$\min_{\pi \in \Pi, \boldsymbol{\xi}} f^i(\pi) + \sigma g(\boldsymbol{\xi}) \tag{64a}$$

$$\text{s.t.} \quad \mathcal{L}_{c_k}^{\pi^i}(\pi) + \xi_k = 0, \quad k = 1, \dots, K \tag{64b}$$

The first term of the problem $f^i(\pi)$ is the linear function of the $\pi$, and the second term is the upper bound of a convex function (KL divergence is the convex function). The function $f^i(\pi)$ is a closed convex function since the linear function is a convex function and finding the upper bound is a convexity operation. By definition, the indication function $g(\boldsymbol{\xi})$ is also a closed-convex function. Constraints are linear functions of policy $\pi$ and relaxation variables $\boldsymbol{\xi}$. If Slater's condition holds, the sequence $\{\pi^{i,j}\}_{j=0}^\infty$ generated by (7a - 7c) converges to $\pi^{i,*}$ can be proven by Boyd et al. (2011).

### A.5 Proof of Theorem 3

**Theorem 3.** *Given the prior estimation $\boldsymbol{\lambda}^{i,j}$ and $\boldsymbol{\xi}^{i,j}$, the optimal policy $\pi^{i,j+1}$ for problem (7a) takes the form:*

$$\pi^{i,j+1}(a|s) = \frac{\pi^i(a|s)}{Z(s)}\left\{\frac{A_{\pi^i}(s,a) - \sum_{k=1}^K[\lambda_k^{i,j} + \rho(\mathcal{L}_{c_k}^{\pi^i}(\pi) + \xi_k^{i,j})]A_{\pi^i}^{c_k}(s,a)}{\eta^{i,j+1}(1-\gamma)}\right\} \tag{65}$$

*where $Z(s)$ serves as a constant normalizer that ensures $\pi$ belongs to the policy set $\Pi$ and the dual variables $\eta^{i,j+1}$ represent the solutions to the following convex optimization problem:*

$$\eta^{i,j+1} = \max_{\eta \geq 0} L(\pi^{i,j+1}, \boldsymbol{\lambda}^{i,j}, \eta, \boldsymbol{\xi}^{i,j}; \sigma, \rho) \tag{66}$$

*Proof.* In order to ensure that the obtained policy $\pi^{i+1}$ belongs to the policy space $\Pi$, here we add the policy space constraint:

$$\pi^{i,j+1} = \arg\min_\pi \ f^i(\pi) + \frac{\rho}{2}\sum_{k=1}^K||\mathcal{L}_{c_k}^{\pi^i}(\pi) + \xi_k^{i,j} + \frac{\lambda_k^{i,j}}{\rho}||_2^2$$

$$\text{s.t.} \quad \sum_{a \in \mathcal{A}} \pi(a|s) = 1, \quad \forall s \in \mathcal{S} \tag{67}$$

The Lagrangian function can be written as follows:

$$\mathcal{F}(\pi, \boldsymbol{\lambda}^{i,j}, \eta, \boldsymbol{\xi}^{i,j}) = -(1-\gamma)^{-1}\mathbb{E}_{s \sim d_{\pi_{\boldsymbol{\theta}^i}}^{\rho_0}, a \sim \pi}[A_{\pi_{\boldsymbol{\theta}^i}}(s,a)] + \eta(D_{KL}(\pi||\pi_{\boldsymbol{\theta}^i}) - \epsilon)$$

$$+ \frac{\rho}{2}\sum_{k=1}^{K}||\mathcal{L}_{c_k}^{\pi^i}(\pi) + \xi_k^{i,j} + \frac{\lambda_k^{i,j}}{\rho}||_2^2 + \sum_{s \in \mathcal{S}}\kappa_s\left(1 - \sum_{a \in \mathcal{A}}\pi(a|s)\right) \qquad (68)$$

Take the derivative of Lagrangian function w.r.t $\pi(a|s)$:

$$\frac{\partial \mathcal{F}}{\partial \pi(a|s)} = -(1-\gamma)^{-1}A_{\pi_{\boldsymbol{\theta}^i}}(s,a) + \eta + \eta\log\frac{\pi(a|s)}{\pi^i(a|s)} + \kappa_s$$

$$+ \rho(1-\gamma)^{-1}\sum_{k=1}^{K}(\mathcal{L}_{c_k}^{\pi^i}(\pi) + \xi_k^{i,j} + \frac{\lambda_k^{i,j}}{\rho})A_{\pi^i}^{c_k}(s,a) \qquad (69)$$

According to the KKT condition, the optimal policy is obtained if and only if equation 69 equals zero, and we have the form of the optimal policy:

$$\pi^{j+1}(a|s) =$$

$$\pi^i(a|s)\exp\{-\frac{\eta^{i,j+1} + \kappa_s}{\eta^{i,j+1}(1-\gamma)}\}\exp\left\{\frac{A_{\pi^i}(s,a) - \sum_{k=1}^{K}[\lambda_k^{i,j} + \rho(\mathcal{L}_{c_k}^{\pi^i}(\pi) + \xi_k^{i,j})]A_{\pi^i}^{c_k}(s,a)}{\eta^{i,j+1}(1-\gamma)}\right\}$$

$$(70)$$

where $\exp\{-\frac{\eta^{i,j+1} + \kappa_s}{\eta^{i,j+1}(1-\gamma)}\}$ is a normalizer of policy $\pi^{j+1}$ to ensure that it is in the policy set $\Pi$.

Take the optimal policy $\pi^{j+1}$ back to the augmented Lagrangian function $L(\pi^{i,j+1}, \boldsymbol{\lambda}^{i,j}, \eta, \boldsymbol{\xi}^{i,j}; \sigma, \rho)$, we can obtain optimal dual variables that can be calculated by

$$\eta^{i,j+1} = \max_{\eta \geq 0} L(\pi^{i,j+1}, \boldsymbol{\lambda}^{i,j}, \eta, \boldsymbol{\xi}^{i,j}; \sigma, \rho) \qquad (71)$$

# B OPTIMIZATION FUNDAMENTALS

## B.1 AUGMENTED LAGRANGIAN

This chapter will introduce some basic knowledge of augmented Lagrangian, mainly considering the following general constraint problems:

$$\min_{\boldsymbol{x}} \quad y(\boldsymbol{x}) \qquad (72a)$$

$$s.t. \quad h_i(\boldsymbol{x}) = 0, i = 1, \ldots, m \qquad (72b)$$

The Lagrangian function form of problem (72) can be expressed as:

$$\min_{\boldsymbol{x}}\max_{\boldsymbol{\lambda}} y(\boldsymbol{x}) + \sum_{i=1}^{m}\lambda_i h_i(\boldsymbol{x}) \qquad (73)$$

It is evident that unless $h_i(\boldsymbol{x}) = 0$, the Lagrange multiplier $\lambda_i$ will tend toward infinity. The optimization of Lagrangian functions becomes particularly challenging due to the inherent lack of smoothness in the maximization operation. The augmented Lagrangian is a method that orchestrates the optimization process to be smoother without altering the optimal solution. The core concept involves incorporating a prior estimate of the Lagrange multipliers and introducing a proximal penalty term to enhance the problem's smoothness. Assuming the existence of a prior estimate $\overline{\boldsymbol{\lambda}}$ for the Lagrangian multiplier, problem (72) can be modified as follows:

$$\min_{\boldsymbol{x}}\left\{\max_{\boldsymbol{\lambda}} y(\boldsymbol{x}) + \sum_{i=1}^{m}\lambda_i h_i(\boldsymbol{x}) - \frac{1}{2\rho}||\boldsymbol{\lambda} - \overline{\boldsymbol{\lambda}}||^2\right\} \qquad (74)$$

In this way, we transform the maximization problem into a quadratic optimization problem, and using Newton's method to optimize, we can obtain update formula of $\lambda$ as:

$$\lambda_i = \overline{\lambda}_i + \rho h_i(\boldsymbol{x}) \tag{75}$$

Substituting equation (75) into problem (74) results in:

$$\min_{\boldsymbol{x}} y(\boldsymbol{x}) + \sum_{i=1}^{m} \overline{\lambda}_i h_i(\boldsymbol{x}) + \frac{\rho}{2} \sum_{i=1}^{m} ||h_i(\boldsymbol{x})||^2 \tag{76}$$

In this way, we obtain the augmented Lagrangian function for the problem (74):

$$L(\boldsymbol{x}, \boldsymbol{\lambda}; \rho) := y(\boldsymbol{x}) + \sum_{i=1}^{m} \lambda_i h_i(\boldsymbol{x}) + \frac{\rho}{2} \sum_{i=1}^{m} ||h_i(\boldsymbol{x})||^2 \tag{77}$$

## B.2 ADMM ALGORITHM

In this chapter, we present a fundamental understanding of the Alternating Direction Method of Multipliers (ADMM) algorithm. We will focus on its application to general constraint problems of the form:

$$\min_{\boldsymbol{x}, \boldsymbol{z}} \quad y(\boldsymbol{x}) + k(\boldsymbol{z}) \tag{78a}$$

$$s.t. \quad A\boldsymbol{x} + B\boldsymbol{z} = \boldsymbol{d} \tag{78b}$$

This optimization problem involves two variables, $\boldsymbol{x}$ and $\boldsymbol{z}$, and a linear equality constraint involving matrices $A$ and $B$, and vectors $\boldsymbol{d}$.

To tackle the problem, we introduce the augmented Lagrangian function:

$$L(\boldsymbol{x}, \boldsymbol{z}, \boldsymbol{\lambda}; \rho) := y(\boldsymbol{x}) + k(\boldsymbol{z}) + \boldsymbol{\lambda}^T(A\boldsymbol{x} + B\boldsymbol{z} - d) + \frac{\rho}{2}||(A\boldsymbol{x} + B\boldsymbol{z} - \boldsymbol{d})||^2 \tag{79}$$

Here, $\boldsymbol{\lambda}$ is the Lagrange multiplier, and $\rho$ is a positive scalar parameter.

The ADMM algorithm iteratively updates the optimization variables $\boldsymbol{x}$, $\boldsymbol{z}$, and the Lagrange multiplier $\boldsymbol{\lambda}$. The updates are as follows:

$$\boldsymbol{x}^{k+1} = \arg\min_{\boldsymbol{x}} \ L(\boldsymbol{x}, \boldsymbol{z}^k, \boldsymbol{\lambda}^k; \rho) \tag{80}$$

$$\boldsymbol{z}^{k+1} = \arg\min_{\boldsymbol{z}} \ L(\boldsymbol{x}^{k+1}, \boldsymbol{z}, \boldsymbol{\lambda}^k; \rho) \tag{81}$$

$$\boldsymbol{\lambda}^{k+1} = \boldsymbol{\lambda}^k + \rho(A\boldsymbol{x}^{k+1} + B\boldsymbol{z}^{k+1} - \boldsymbol{d}) \tag{82}$$

The algorithm iteratively refines the solutions for $\boldsymbol{x}$ and $\boldsymbol{z}$ while updating the Lagrange multiplier $\boldsymbol{\lambda}$ to enforce the constraint.

In addition to the standard ADMM form mentioned above, a more concise variant, known as Scaled Form ADMM, has been developed. This variant involves modifications that can lead to more efficient convergence. Specifically, the optimization approach for Scaled Form ADMM involves:

$$\boldsymbol{x}^{k+1} = \arg\min_{\boldsymbol{x}} \ y(\boldsymbol{x}) + \frac{\rho}{2}||A\boldsymbol{x} + B\boldsymbol{z}^k - \boldsymbol{d} + \frac{\boldsymbol{\lambda}^k}{\rho}||^2 \tag{83}$$

$$\boldsymbol{z}^{k+1} = \arg\min_{\boldsymbol{z}} \ k(\boldsymbol{z}) + \frac{\rho}{2}||A\boldsymbol{x}^{k+1} + B\boldsymbol{z} - \boldsymbol{d} + \frac{\boldsymbol{\lambda}^k}{\rho}||^2 \tag{84}$$

$$\boldsymbol{\lambda}^{k+1} = \boldsymbol{\lambda}^k + \rho(A\boldsymbol{x}^{k+1} + B\boldsymbol{z}^{k+1} - \boldsymbol{d}) \tag{85}$$

To demonstrate the equivalence relationship between optimization problem (80) and problem (83), we provide the following analysis:

$$\boldsymbol{x}^{k+1} = \arg\min_{\boldsymbol{x}} \; L(\boldsymbol{x}, \boldsymbol{z}^k, \boldsymbol{\lambda}^k; \rho) \tag{86}$$

$$= \arg\min_{\boldsymbol{x}} \; y(\boldsymbol{x}) + k(\boldsymbol{z}^k) + (\boldsymbol{\lambda}^k)^T(A\boldsymbol{x} + B\boldsymbol{z}^k - d) + \frac{\rho}{2}||(A\boldsymbol{x} + B\boldsymbol{z}^k - \boldsymbol{d})||^2 \tag{87}$$

$$= \arg\min_{\boldsymbol{x}} \; y(\boldsymbol{x}) + \frac{\rho}{2}||\frac{\boldsymbol{\lambda}^k}{\rho}||^2$$

$$+ (\boldsymbol{\lambda}^k)^T(A\boldsymbol{x} + B\boldsymbol{z}^k - d) + \frac{\rho}{2}||(A\boldsymbol{x} + B\boldsymbol{z}^k - \boldsymbol{d})||^2 - \frac{\rho}{2}||\frac{\boldsymbol{\lambda}^k}{\rho}||^2 \tag{88}$$

$$= \arg\min_{\boldsymbol{x}} \; y(\boldsymbol{x}) + \frac{\rho}{2}||A\boldsymbol{x} + B\boldsymbol{z}^k - \boldsymbol{d} + \frac{\boldsymbol{\lambda}^k}{\rho}||^2 \tag{89}$$

Similar methodologies can lead to:

$$\boldsymbol{z}^{k+1} = \arg\min_{\boldsymbol{z}} \; k(\boldsymbol{z}) + \frac{\rho}{2}||A\boldsymbol{x}^{k+1} + B\boldsymbol{z} - \boldsymbol{d} + \frac{\boldsymbol{\lambda}^k}{\rho}||^2 \tag{90}$$

## C EXPERIMENT ENVIRONMENTS

### C.1 SAFE VELOCITY

Velocity tasks are a significant category of real-world applications, where agents must maximize their speed while adhering to velocity constraints. In this task, a safety constraint is imposed on the agent's speed in Mujoco. We utilize four unique agents, each with its own configuration: *SafetyAnt*, *SafetyHalfCheetah*, *SafetySwimmer* and *SafetyHumanoid* as illustrated in Figure 4. The reward function $R_t$ comprises three components, though certain agents may disregard some of them:

- $r_{healthy}$: Every timestep that the agent is healthy, it gets a reward of fixed value $r_{healthy}$.
- $r_{forward}$: A reward of moving forward which is measured as $(x_t - x_{t-1})/dt$. $dt$ is the time step. $x_t$ is the x-coordinate after action and $x_{t-1}$ is the x-coordinate before action. This reward would be positive if the ant moves forward (in the positive x direction).
- $r_{ctrl}$: A negative reward for penalizing the ant if it takes actions that are too large. It is measured as $w_{ctrl} \cdot \sum action^2$ where $w_{ctrl}$ is a parameter set for the control and has a default value of 0.5.

The total reward returned is $R_t = r_{healthy} + r_{forward} - r_{ctrl}$. And the cost function $C_t$ can be formulated as follows:

$$C_t = \mathbb{I}(V_{current} > V_{threhold})$$

where $V_{current}$ represents the current speed of the agent and $V_{threhold}$ is the threshold speed, which is set to a maximum of 50% of the agent's speed after one million steps of optimization using PPO algorithms. For more comprehensive information on the environment and agent settings, please refer to: https://www.safety-gymnasium.com/en/latest/environments/safe_velocity.html

### C.2 SAFE NAVIGATION

#### C.2.1 CIRCLE TASK

In the *Circle*task, the agent's goal is to navigate a circular area while optimizing speed and maintaining distance from the center to receive rewards. However, venturing into unsafe zones results in penalties. We utilize two agent types, referred to as the *Point* and *Car* agents, as depicted in Figure 5. The task environment is illustrated in Figure 6(a). The observation is a vector that comprises

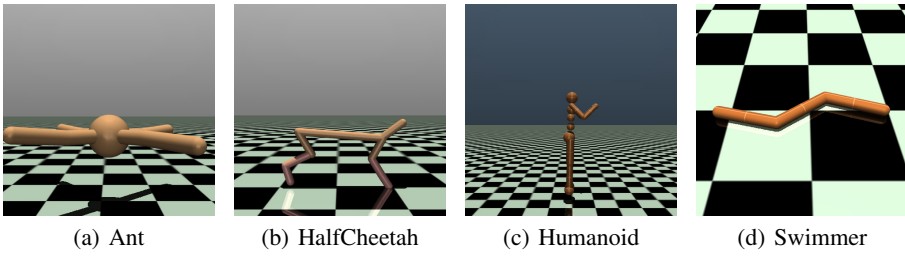

(a) Ant      (b) HalfCheetah      (c) Humanoid      (d) Swimmer

Figure 4: Agent in safe velocity

information about the agent's current state and pseudo LIDAR points. The reward function $R_t$ and cost function $C_t$ are defined as follows:

$$R_t = \frac{1}{1 + |r_{taget} - r_{circle}|} * \frac{-(uy + vx)}{r_{agent}}$$
$$C_t = \mathbb{I}(|x| > x_{limit})$$

where $u$ and $v$ represent the velocity components of the agent along the x and y axes, while $x$ and $y$ denote the agent's coordinates in the x and y axes. $r_{agent}$ represents the Euclidean distance of the agent from the origin, and $r_{circle}$ corresponds to the radius of the circle geometry. $\mathbb{I}()$ represents the indicator function and $x_{limit}$ is the safety margin. For more comprehensive information on the environment and agent settings, please refer to: https://www.safety-gymnasium.com/en/latest/environments/safe_navigation/circle.html

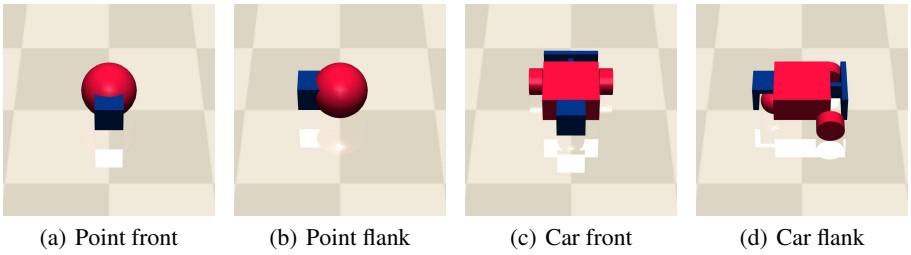

(a) Point front      (b) Point flank      (c) Car front      (d) Car flank

Figure 5: Agent in safe navigation

### C.2.2 GOAL TASK

In *Goal* task, the agent is tasked with reaching a series of goal positions. Upon reaching a goal, the goal location is randomly reset to a new position, while maintaining the same environment layout. The sparse reward component is earned when the robot successfully reaches a goal position (enters the goal circle), while the dense reward component provides a bonus for making progress toward the goal. Observation is the same as the *Circle* task environment. The task environment is illustrated in Figure 6(b). The reward function $R_t$ and cost function $C_t$ are defined as follows:

$$R_t = (D_{last} - D_{now})\beta + R_{goal}\mathbb{I}(D_{now} < D_{threhold})$$
$$C_t = C_{touch} \max\{S_{agent} - D_{hazard}\}$$

$D_{last}$ represents the distance between the agent and the target point at the previous time step, $D_{now}$ represents the distance between the robot and the target point at the current time step. $\beta$ is the discount factor, and $R_{goal}$ is a positive value that represents the reward for completing a goal when it is reached. $D_{threshold}$ is the threshold for judgment, $S_{agent}$ is the size of the agent, $D_{hazard}$ is the distance between the agent and a hazard and $C_{touch}$ is the penalty per unit time . For more comprehensive information on the environment and agent settings, please refer to: https://www.safety-gymnasium.com/en/latest/environments/safe_navigation/goal.html

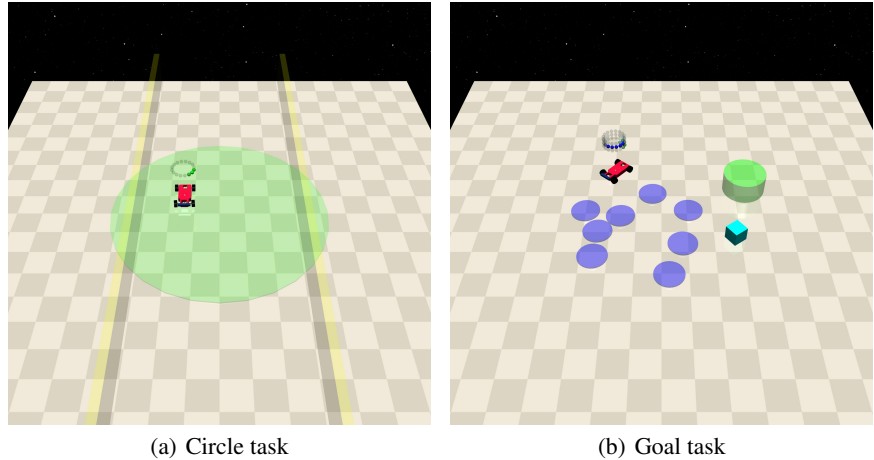

(a) Circle task          (b) Goal task

Figure 6: Safe navigation

## D ALGORITHM DETAILS

Upon observation, we identified that the variables $\xi^{i,j+1,*}$ in the optimization problems described above can be effectively decoupled. Consequently, we can derive the optimal value of $\xi_k^{i,j+1,*}$ by solving the following optimization problem:

$$\min_{\xi_k} \left\{ \sigma\max\{0, -\xi_k\} + \frac{\rho}{2}||\mathcal{L}_{c_k}^{\pi^i}(\pi^{i,j+1}) + \xi_k + \frac{\lambda_k^{i,j}}{\rho}||_2^2 \right\} \tag{91}$$

If we make the assumption that $\xi_k^{i,j+1,*} > 0$, we can establish the condition for it to satisfy a gradient of 0 as follows:

$$\rho(\mathcal{L}_{c_k}^{\pi^i}(\pi^{i,j+1}) + \xi_k^{i,j+1,*} + \frac{\lambda_k^{i,j}}{\rho}) = 0 \tag{92}$$

$$\xi_k^{i,j+1,*} = -\mathcal{L}_{c_k}^{\pi^i}(\pi^{i,j+1}) - \frac{\lambda_k^{i,j}}{\rho} \tag{93}$$

If we make the assumption that $\xi_k^{i,j+1,*} \leq 0$, we can establish the condition for it to satisfy:

$$-\sigma + \rho(\mathcal{L}_{c_k}^{\pi^i}(\pi^{i,j+1}) + \xi_k^{i,j+1,*} + \frac{\lambda_k^{i,j}}{\rho}) \leq 0 \tag{94}$$

$$\xi_k^{i,j+1,*} \leq \frac{\sigma}{\rho} - \mathcal{L}_{c_k}^{\pi^i}(\pi^{i,j+1}) - \frac{\lambda_k^{i,j}}{\rho} \tag{95}$$

In summary, we can obtain:

$$\xi_k^{i,j+1,*} = \begin{cases} -\mathcal{L}_{c_k}^{\pi^i}(\pi^{i,j+1}) - \frac{\lambda_k^{i,j}}{\rho}, & -\mathcal{L}_{c_k}^{\pi^i}(\pi^{i,j+1}) - \frac{\lambda_k^{i,j}}{\rho} > 0 \\ \min\{0, \frac{\sigma}{\rho} - \mathcal{L}_{c_k}^{\pi^i}(\pi^{i,j+1}) - \frac{\lambda_k^{i,j}}{\rho}\}, & \text{otherwise} \end{cases} \tag{96}$$

To enhance the algorithm's convergence stability in the presence of sampling errors, we have made modifications to the update process as follows:

$$\xi_k^{i,j+1} = (1 - \vartheta)\xi_k^{i,j} + \vartheta\xi_k^{i,j+1,*} \tag{97}$$

where $\vartheta \in [0, 1]$ is the update weight.

In accordance with the principles of ADMM, $\lambda$ achieves the fastest convergence when its learning rate matches $\rho$, yet a smaller learning rate contributes to algorithm stability. Consequently, in practical algorithms, we employ the following formula to update $\lambda$:

$$\lambda_k^{i,j+1} = \lambda_k^{i,j} + \varrho(\mathcal{L}_{c_k}^{\pi^i}(\pi^{i,j+1}) + \xi_k^{i,j+1}) \tag{98}$$

where $\varrho$ is the learning rate.

Additionally, to attain superior convergence results, we implement a warm-up technique for penalty coefficients $\sigma$. The specific implementation of ACPO can be found in Algorithm 3.

---

**Algorithm 3** Augmented Constraint Policy Optimization (ACPO)

---

**Input:** Policy network $\pi_{\boldsymbol{\theta}}$, Value networks $V_{\boldsymbol{\phi}}, V_{\boldsymbol{\phi}_1}^{c_1}, \ldots, V_{\boldsymbol{\phi}_K}^{c_K}$, Discount rates $\gamma$, GAE parameter $\beta$, Learning rates $\alpha_V, \alpha_\pi, \vartheta, \varrho$, Penalty parameter $\sigma_{max}$, Warmup rate $\Delta$, Trust region bound $\epsilon$, Cost bound $b_1, \ldots, b_K$.

**while** Stopping criteria not met **do**

    Generate dataset $\mathcal{D} = \{s_{m,t}, a_{m,t}, r_{m,t}, s_{m,t+1} \ldots, c_{m,t}^k\}$ of $M$ episodes of length $T$ from $\pi_{\boldsymbol{\theta}}$.

    Estimate $k$-th return by:

$$\hat{J}_k = \frac{1}{M} \sum_{m=1}^{M} \sum_{t=0}^{T-1} \gamma^t c_{m,t}^k$$

    Estimate advantage functions: $\hat{A}(s_{m,t}, a_{m,t}), \ldots, \hat{A}^{c_K}(s_{m,t}, a_{m,t})$ using GAE.

    Estimate value functions:

    $V^t(s_{m,t}) = \hat{A}(s_{m,t}, a_{m,t}) + V_{\boldsymbol{\phi}}(s_{m,t}), \ldots, V^{c_K,t}(s_{m,t}) = \hat{A}(s_{m,t}, a_{m,t}) + V_{\boldsymbol{\phi}_K}^{c_K}(s_{m,t})$

    Store old policy $\boldsymbol{\theta}' \leftarrow \boldsymbol{\theta}$.

    **for** each iteration **do**

        **for** each minibatch of size $B$ **do**

            Get value loss functions:

$$\mathcal{L}_V(\boldsymbol{\phi}) = \frac{1}{2B} \sum_{b=1}^{B} (V_{\boldsymbol{\phi}}(s_b) - V^t(s_b))^2$$

$$\mathcal{L}_{V_k}(\boldsymbol{\phi}_k) = \frac{1}{2B} \sum_{b=1}^{B} (V_{\boldsymbol{\phi}_k}^{c_k}(s_b) - V^{c_k,t}(s_b))^2, k = 1, \ldots, K$$

            Update value networks:

$$\boldsymbol{\phi} \leftarrow \boldsymbol{\phi} - \alpha_V \nabla \mathcal{L}_V(\boldsymbol{\phi})$$
$$\boldsymbol{\phi}_k \leftarrow \boldsymbol{\phi}_k - \alpha_V \nabla \mathcal{L}_{V_k}(\boldsymbol{\phi}_k), k = 1, \ldots, K$$

            Update policy network:

$$\boldsymbol{\theta} \leftarrow \boldsymbol{\theta} - \alpha_\pi \nabla Loss(\boldsymbol{\theta})$$

        **end for**

        **if** $\frac{1}{MT} \sum_{i=1}^{M} \sum_{t=0}^{T} D_{KL}(\pi_{\boldsymbol{\theta}} || \pi_{\boldsymbol{\theta}'})[s_{m,t}] > \epsilon$ **then**

            Break

        **end if**

        Calculate the approximation function:

$$\hat{\mathcal{L}}_k = \hat{J}_k + \frac{1}{MT} \sum_{m=1}^{M} \sum_{t=0}^{T} [\hat{A}(s_{m,t}, a_{m,t})] - b_k, \quad k = 1, \ldots, K$$

        Update $\boldsymbol{\xi}$:

$$\xi_k^* = \begin{cases} -\hat{\mathcal{L}}_k - \frac{\lambda_k}{\rho}, & -\hat{\mathcal{L}}_k - \frac{\lambda_k}{\rho} > 0 \\ \min\{0, \frac{\sigma}{\rho} - \hat{\mathcal{L}}_k - \frac{\lambda_k}{\rho}\}, & \text{otherwise} \end{cases}$$

$$\xi_k \leftarrow (1 - \vartheta)\xi_k + \vartheta \xi_k^*, \quad k = 1, \ldots, K$$

        Update $\boldsymbol{\lambda}$:

$$\lambda_k \leftarrow \lambda_k + \varrho(\hat{\mathcal{L}}_k + \xi_k), \quad k = 1, \ldots, K$$

    **end for**

    Warmup the penalty parameter $\sigma \leftarrow \min\{\sigma_{max}, \sigma + \Delta\}$

**end while**

---

