# OpenReview forum: "Augmented Policy Optimization for Safe Reinforcement Learning"
_ICLR.cc/2024/Conference — ICLR 2024 Conference Withdrawn Submission_

### Official Review · Reviewer_Q99o · 2023-10-19

**Soundness:** 2 fair
**Presentation:** 3 good
**Contribution:** 2 fair
**Rating:** 3
**Confidence:** 4

**Summary:**

The paper proposes ACPO algorithm to solve a safe RL problem. ACPO first converts the objective in CPO to a penalized one and then solve the augmented Lagrangian problem by ADMM. The authors conduct the experiments on tasks in Safety-gymnasium and compare the proposed method to previous baselines.

**Strengths:**

1. Safe RL is a topic of significant interest to the community and active research.
2. The paper is clearly written and easy to follow.

**Weaknesses:**

1. The theoretical contribution of this paper is very incremental. The key upper bound in proposition 1 and theorem 2 comes from CPO. The closed-form solution of optimal policy in theorem 3 resembles FOCOPS. In fact, the provided theoretical results do not explain how the proposed method exceeds the baselines in terms of final performance or training stability.
2. In the experiment, the proposed method obtains very similar performances to baselines. In fig.2, ACPO has no remarkable advantage over CPPOPID and FOCOPS in terms of both reward improvement and constraint satisfaction. In fig.3, ACPO cannot even converge to satisfy the cost constraint after 10M steps on Goal tasks -- the hardest tasks in this paper. Although ACPO combines the components from several previous methods, there is no significant empirical improvement over them.
3. Although this paper roots on the instability issue of previous safe RL method, the proposed method does not show better stability. It can be found the variance scale of ACPO is similar to FOCOPS in table 3, and it also suffers from early-stage cost fluctuations in fig.3.

**Questions:**

1. Should the term $A_\pi$ in second line of eq.(10) be $A_{\pi}^c$ instead? Otherwise, the policy update will not consider any term related to cost.
2. Could you elaborate the setting of "generalization analysis" in the experiment? Do you use the same random seed for training and test? If they are different, why are the performances on "generalization" setting superior to them in original setting? E.g., ACPO achieves 94 rewards on PointCircle in table 1 but it's only around 45 in figure 3.
3. Why do you use warmup for $\sigma$? Is it used to encourage early-stage exploration?

---

### Official Review · Reviewer_Bx7v · 2023-10-21

**Soundness:** 3 good
**Presentation:** 3 good
**Contribution:** 3 good
**Rating:** 6
**Confidence:** 4

**Summary:**

This paper deals with so-called safe reinforcement learning (RL) problems. To address the issues in the existing primal-dual or trust-region methods, the authors propose an algorithm called Augmented Constraint Policy Optimization (ACPO), which encompasses a novel approach to constructing local policy search problems and an optimization problem decomposition method. Empirically, the authors demonstrated that their proposed method performs better than the reasonable baselines including SOTA methods in terms of both performance and constraint satisfaction.

**Strengths:**

- This paper is well-written and organized. Existing work on safe RL is widely cited and it is easy for readers to catch up the related work in this field.
- The proposed algorithm is technically-sound. The intuitive explanations are properly conducted and it is easy to understand how the algorithm works.
- Theoretical analysis is well-conducted. Though I feel that theoretical results are largely based on the existing work (e.g., Achiam+ (2017) or Boyd+ (2011)), I do not consider that it is quite problematic.
It is because this paper is not purely theoretical and there are indeed some advancements in terms of theoretical analysis.
- The empirical experiments are also good. Though I have some concerns mentioned below, the empirical evaluation of this paper is overall well-conducted.

**Weaknesses:**

- The authors discuss projection-based approaches such as Yang (2020), but this previous work seems to be omitted from the baseline methods. This method is a quite relevant method as discussed by the authors, I think that it would be better to include it as a baseline.
- Though the authors emphasize that the proposed algorithm is stable, I am not fully convinced whether it is actually stable depending on the initial settings of the parameters. For example, in (10), there are so many parameters, at least I do not have confidence to successfully optimize a policy and each parameter.  I understand if each parameter is set to a proper value, then the policy is learned in a stable manner. Regarding this, I ask a question below.
- Though it is quite recent, there is the following paper that is strongly related. I think it is better to cite it and discuss their differences at least "in text" (I do not require the authors to conduct experiments since this paper can be viewed as a concurrent work). Also, the title is almost same and the only difference is with or without "Proximal". I would recommend the authors to change the title in order to make the difference clearer.
    - Dai et al., "Augmented Proximal Policy Optimization for Safe Reinforcement Learning". https://ojs.aaai.org/index.php/AAAI/article/view/25888
- Regarding the experiments, I do not think that authors provide the details for reproducing their experiments. Unless I miss something, there is no source-code attached and no table for the parameters used in their experiments. I think it is almost impossible to reproduce the authors' results from the paper.

**Minor Comments**

- When I first read this paper, Figure 1 was unclear. The authors may want to represent what is blue region in (b).

**Questions:**

[Q1] While I understand that the authors made much effort to develop a stable algorithm, I feel that there are many parameters (hyper-parameters, Lagrange multiplier, etc.) that must be tuned, which results in an unstable algorithm as a whole process of policy learning. Could you comment on my concerns?

[Q2] Is the necessary assumption only the Slater condition? If there are other assumptions, I would recommend the authors to explicitly note it in the paper.

[Q3] Regarding $g(\xi^i)$, I think this function is not differentiable. Practically, is this approximated by some differentiable function such as $\log(\cdot)/\alpha$ as conducted in IPO?

Note that, though my current rating is "6: marginally above the acceptance threshold", I would happy to change the score based on the authors' rebuttal and revision since I consider that this paper is substantially nice.

---

### Official Review · Reviewer_3HXh · 2023-10-22

**Soundness:** 3 good
**Presentation:** 3 good
**Contribution:** 2 fair
**Rating:** 5
**Confidence:** 3

**Summary:**

In this paper, the authors delve into the Safe Reinforcement Learning (Safe RL) problem, proposing the Augmented Constraint Policy Optimization (ACPO) algorithm. This novel approach employs local policy search problems and an optimization problem decomposition method, ensuring efficient and robust constrained RL policy updates. The proposed method is validated through theoretical analysis and comprehensive experiments.

**Strengths:**

(1) Significance of the problem: The study addresses a critical issue in real-world applications.

(2) Extensive experimental validation: The authors back their proposal with an extensive set of experiments.

(3) Theoretical foundation: The inclusion of theoretical analysis strengthens the credibility of the proposed method.

**Weaknesses:**

(1) Unavailable code link: Considering the importance of practical implementation, it would greatly benefit the readers if the authors could provide access to the code repository.

(2) Lack of experiment details: A more comprehensive discussion of the experiments, including a breakdown of evaluation metrics and insights into why ACPO demonstrates strong generalizability, would enhance the clarity of the findings.

(3) Ambiguous conclusion statement: The authors claim that since the algorithm processes decomposition characteristics, it is well-suited for expansion for multi-agent systems. Clarification is needed regarding the complexity of the proposed method in relation to multi-agent systems. It would be helpful to know if the proposed method incurs high sampling or computational complexity and to understand the implications for multi-agent systems.

(4) Confusing experiment results: The safety performance of ACPO is not good enough. (See my question 5)

**Questions:**

(1) Practical implementation: Could you elaborate on how the penalty coefficient sigma in Theorem 2 is selected?

(2) Comparison with related methods: It would be beneficial if the authors discussed similar methods that tackle the conversion of hard safety constraints into soft constraints.

(3) Definition clarification: What does “security RL” refer to in the conclusion?

(4) Adaptability and comparisons: Can you elaborate on the adaptability of the proposed method to off-policy methods and whether any comparisons were made with relevant off-policy baselines?

(5) Safety performance discussion: Could you provide further insights into the safety performance depicted in Figures 2 and 3, especially in instances where the ACPO method struggles to converge to the target cost threshold in tasks like Safety-Swimmer, CarGoal, and PointGoal?

---

### Official Review · Reviewer_wA8B · 2023-11-01

**Soundness:** 2 fair
**Presentation:** 3 good
**Contribution:** 2 fair
**Rating:** 3
**Confidence:** 4

**Summary:**

This paper proposed a local policy search method for a constrained Markov decision process. When the problem is infeasible, the local policy search might have no solution. The authors augmented one penalty term to soften the objective function so that the local policy search is always feasible. Moreover, an ADMM-based algorithm is proposed to solve the local policy search. Theoretical results claim that the optimal solution given known dual variables and convergence to the optimal policy. Empirical results verifies the effectiveness of the proposed algorithm.

**Strengths:**

1. The presentation is good, the problem formulation and algorithms are presented clearly.
2. Applying ADMM on local policy search looks novel to me.

**Weaknesses:**

1. The paper is of minor contributions. Using slack variables to soften constraints is commonly used in handling infeasibility issues of optimizations. The motivations for applying ADMM to local policy search are not clear.
2. There might be technical flaws in the paper. The theoretical results might not be correct. For example, Lemma 1 might not be correct. The actual decision variable is not policy $\pi$, it is the parameters of the policy network. The neural network parameterization is usually nonconvex, so the strong duality usually does not hold.
3. There are a lot of statements in the paper that are careless and not technically correct. I doubt the authors might not have enough knowledge of the related fields. For example, in section 2.2, *a policy π is a probability distribution defined on $\mathcal S\times\mathcal A$*, it is not. After lemma 1, the authors claim that the algorithm will ultimately *achieve* convergence, the theoretical results do not provide finite-time convergence, so it's only asymptotic convergence, you might not say you actually "achieve" the convergence.
4. The performance in the empirical results does not outperform the baselines.

**Questions:**

1. What are the motivations of using ADMM for solving local policy search? If it is from the common ADMM benefit, such as the computational efficiency, please give a comparison of the computations between ADMM and the baselines.
2. Please explain convergence proof in Lemma 1 mentioned in weakness 2, which is the major technical problem in this paper.